# Mixed-methods feasibility study of blood pressure self-screening for hypertension detection

Alice Tompson,[1] Susannah Fleming,[1] Mei-Man Lee,[1] Mark Monahan,[2] Sue Jowett,[2] David McCartney,[1] Sheila Greenfield,[3] Carl Heneghan,[1] Alison Ward,[1] Richard Hobbs,[1] Richard J McManus[1]

AT and SF contributed equally.

¹Nuffield Department of Primary Care Health Sciences, University of Oxford, Oxford, UK
²Health Economics Unit, University of Birmingham, Birmingham, UK
³Institute of Applied Health Service Research, University of Birmingham, Birmingham, UK

**Correspondence to**
Dr Susannah Fleming;
susannah.fleming@phc.ox.ac.uk

## ABSTRACT

**Objective** To assess the feasibility of using a blood pressure (BP) self-measurement kiosk—a solid-cuff sphygmomanometer combined with technology to integrate the BP readings into patient electronic medical records— to improve hypertension detection.

**Design** A concurrent mixed-methods feasibility study incorporating observational and qualitative interview components.

**Setting** Two English general practitioner (GP) surgeries.

**Participants** Adult patients registered at participating surgeries. Staff working at these sites.

**Interventions** BP self-measurement kiosks were placed in the waiting rooms for a 12-month period between 2015 and 2016 and compared with a 12-month control period prior to installation.

**Outcome measures** (1) The number of patients using the kiosk and agreeing to transfer of their data into their electronic medical records; (2) the cost of using a kiosk compared with GP/practice nurse BP screening; (3) qualitative themes regarding use of the equipment.

**Results** Out of 15 624 eligible patients, only 186 (1.2%, 95% CI 1.0% to 1.4%) successfully used the kiosk to directly transfer a BP reading into their medical record. For a considerable portion of the intervention period, no readings were transferred, possibly indicating technical problems with the transfer link. A comparison of costs suggests that at least 52.6% of eligible patients would need to self-screen in order to bring costs below that of screening by GPs and practice nurses. Qualitative interviews confirmed that both patients and staff experienced technical difficulties, and used alternative methods to enter BP results into the medical record.

**Conclusions** While interviewees were generally positive about checking BP in the waiting room, the electronic transfer system as tested was neither robust, effective nor likely to be a cost-effective approach, thus may not be appropriate for a primary care environment. Since most of the cost of a kiosk system lies in the transfer mechanism, a solid-cuff sphygmomanometer and manual entry of results may be a suitable alternative.

## INTRODUCTION

Hypertension is a major risk factor for adverse cardiovascular events such as stroke and myocardial infarction.[1] Interventions to reduce blood pressure (BP), including both lifestyle modifications such as changes in diet and physical activity levels and antihypertensive medication, can reduce cardiovascular mortality and morbidity in the hypertensive population,[2–6] and are cost-effective.[7 8] However, as raised BP is typically asymptomatic, screening is necessary to detect those requiring a formal diagnostic process for hypertension and subsequently provide them with effective management.[9 10]

In the UK, BP screening has typically been carried out by clinicians in an ad hoc manner—monitoring patients who may be presenting for other reasons.[11] Regular measurement of BP in adults has been included in the National Health Service (NHS) Quality Outcomes Framework since 2004.[12] The introduction of the NHS Health Check in 2008 further incentivised screening of all adults aged 40–74 years, providing the opportunity to capture those who do not require or request routine medical care, and would be missed by opportunistic screening.[13]

The advent of solid-cuff sphygmomanometers[14] has allowed BP screening to move from the consultation into the waiting room.[15] Patients can use these BP monitors without any assistance from staff, and monitors are often equipped with printers, allowing the result to be reliably recorded.

## Strengths and limitations of this study

► Blood pressure (BP) self-screening kiosks offer the potential by which to address the paucity of studies investigating the impact of BP self-screening on hypertension diagnosis.
► Adopting a mixed-methods approach enables a more complete picture of feasibility to be painted.
► The study design relies on quantitative data being present and extractable from electronic medical records.

However, to date, there is still a need for manual intervention from staff to enter the BP reading into the patient's electronic medical record and interpret it.

A number of manufacturers now offer kiosks that combine a solid-cuff sphygmomanometer with a computer interface, allowing the patient to identify themselves with links to the local clinical system, so that any BP readings are automatically added to the patient's medical record. These kiosks typically include the option to set thresholds to notify clinicians of BP readings that are outside acceptable ranges, so that patients can be recalled for further tests if necessary. They may also provide the opportunity to carry out other forms of screening such as weight or screening questionnaires (eg, https://www.thsl.co.uk/products/surgerypod-2).

In this study, we assessed the feasibility of using such a kiosk to improve detection of hypertension in the community using a mixed-methods approach.

## METHODS
### Quantitative data
We selected a BP self-measurement kiosk device (SurgeryPod Plus, Medvivo latterly Microtech, UK) that was able to interface with various UK NHS general practice electronic health record systems, and which took seated BP using an clinically validated sphygmomanometer. BP self-measurement kiosks were placed in the waiting room areas of two general practitioner (GP) surgeries in the UK. The practices were selected due to their differing sizes, patient population characteristics and IT systems. The intention was to also include a pharmacy site, but no pharmacy partner who was willing and able to participate could be identified.

The kiosks combined a clinically validated solid-cuff automated sphygmomanometer (TM-2655P, A&D Instruments, UK), supporting arm circumferences from 17 to 45 cm, with a dedicated touch screen computer interface.[14] The touch screen allowed the patient to identify themselves within the practice list, and then guided them through the BP measurement. Patients were asked to take three BP measures within a single session, and were then offered the opportunity to have the readings transferred to their electronic medical record. A measurement session lasted about 4 min. If the BP reading(s) were above the predefined trigger level of 135/85 mm Hg, and the patient had agreed to the transfer of their results, the kiosk software would trigger an email to a nominated practice account. The trigger level was chosen to match the clinical threshold for home and ambulatory BP measurement in the UK National Institute for Health and Care Excellence guidelines.[10]

This commercially available system was chosen over other products because it measured seated BP via a validated device, was UK based and integrated the results into the electronic medical record system enabling quantitative evaluation.

A kiosk was installed in the waiting room area of each of the two participating GP surgeries for a 12-month period with a 12-month control period prior to installation of the kiosk also identified. At the end of the study period, the following anonymised data were extracted from the electronic medical record:

► Demographics (age, gender, ethnicity) of population using the kiosk.
► Hypertension diagnosis dates for patients registered during both the study and control periods.
► All BP recorded during the study and control periods, including those recorded using the kiosk.
► Use of home and ambulatory BP monitoring during the study and control periods.
► All clinician consultations during the study and control periods.

Data were also collected on medications, cardiovascular disease risk scoring, cholesterol blood test results and other comorbidities for use in planned health economic modelling (see below for economic analyses).

The kiosk was installed in the waiting room of practice A from 25 June 2015 to 24 June 2016. The control period identified for this site was 9 May 2013 to 8 May 2014. This was a year earlier than the immediate intervention period due to a typographical error in the search used to extract the anonymised data. The intervention period for practice B was 2 September 2015 to 1 September 2016 and the control period was 2 September 2014 to 1 September 2015.

The manufacturer was able to remotely monitor the number of screening sessions based on data transferred from the kiosk but not the screening data itself. These data were shared with the study team following the completion of the study.

To assess the feasibility of the kiosks, the primary outcome chosen was the number of patients using them and agreeing to transfer of their data into their electronic medical records. Investigation of the demographics of kiosk users, as well as whether they had a prior hypertension diagnosis, and whether they had any clinician measurement of BP during the study period was also planned.

It was hypothesised that use of the kiosks by patients without a hypertension diagnosis might lead to increased detection of undiagnosed hypertension, so rates of hypertension diagnosis in the study and control periods, and in those using and not using the kiosk were assessed. Similarly, the number of those who screened high on the kiosk and subsequently received diagnostic assessment for hypertension (defined as home or ambulatory BP monitoring) was evaluated.

The potential effect of the kiosks on the clinical workload was considered by ascertaining how many people screened high on their kiosk reading, and subsequently attended for a consultation (without any way of knowing if the subsequent consultation was related to the high reading). Comparison was made of the rates of clinician measurement of BP during the study and control

periods, to see if the presence of the kiosk altered clinician screening workload. This was tested using a $X^2$ test with continuity correction.

All numerical analyses were carried out using R V.3.4.2.[16]

## Economic analyses

In the light of the results of the study, a simple economic analysis was undertaken. The analysis compared the cost of a practice using a self-screening kiosk for 1 year, with BP screening undertaken by a combination of GPs and practice nurses (base case) and also with practice nurse only screening and healthcare assistant (HCNA) only screening. Using data from the control period of one of the practices (practice A), it was calculated that 2681 patients (53.5% of the adult practice population) were aged over 45 years with no diagnosed hypertension, and with screening every 5 years, one-fifth (n=536) could potentially be screened in 1 year. The actual cost of hiring a kiosk for 1 year was obtained within the study (see online supplementary table 1). For healthcare professional screening, the cost for a 5 min appointment was obtained for screening by a GP, practice nurse and HCNA (see online supplementary table 1).[17] Five minutes was chosen as the minimum time needed to take a satisfactory BP measurement. The total costs of screening all patients were calculated and compared with the cost of a kiosk. The proportion screened via a kiosk was also varied, assuming that the remaining patients were screened by a healthcare professional. Further modelling was not undertaken.

## Qualitative data

GPs and practice staff were invited to interview and informed consent obtained. The interviews were conducted face to face by a non-clinical researcher (AT) based on a topic guide informed by the evaluation aims. Interview recordings were transcribed and checked. Initial transcripts were coded by two researchers in NVivo and a coding framework discussed. This was applied to the subsequent transcripts and refined. Earlier transcripts were re-examined for codes that appeared later in the dataset.

As part of a linked study investigating patient experience of BP measurement during hypertension diagnosis, a series of interviews were conducted with patients on the hypertension register at the pilot practices (for full details see Tompson *et al*, manuscript submitted). This dataset was re-examined for codes regarding the self-screening and the kiosk. Codes from the two sets of interviews were combined summarised using the one sheet of paper technique.[18]

## Mixed-methods approach

In order to gain a more complete picture of feasibility, both quantitative and qualitative methods were used concurrently to use the strengths and offset the limitations of each approach.[19] Following separate analyses, the multidisciplinary research team triangulated the findings, using qualitative codes from the summarised interview data to illustrate and explore the quantitative findings, and consider the areas in which there was convergence, complementarity or contradiction.[20] Due to the composition of the research team and the biomedical paradigm in which they work, the quantitative data were used as the 'starting point' for corroboration when considering the study findings.

**Table 1** Monitoring data from the kiosk supplier showing the number of kiosk sessions for which data were transferred to the practice system during each month of the study period

| Month | Practice A | | Practice B | |
| | Sessions transferred | Comment | Sessions transferred | Comment |
|---|---|---|---|---|
| 1 | 10 | No activity for 21 days | 40 | |
| 2 | 7 | No activity for 11 days | 30 | |
| 3 | 0 | No activity | 51 | |
| 4 | 34 | | 35 | |
| 5 | 27 | | 16 | |
| 6 | 16 | | 22 | |
| 7 | 19 | | 6 | No activity for 29 days |
| 8 | 0 | No activity | 4 | No activity for 11 days, some activity and then no activity for 11 days |
| 9 | 0 | No activity | 0 | No activity |
| 10 | 0 | No activity | 0 | No activity |
| 11 | 0 | No activity | 0 | No activity |
| 12 | 0 | No activity | 0 | No activity |
| Total | 113 | | 204 | |

The actual dates are different for each site due to different intervention start dates.

**Table 2** Characteristics of patients who used a blood pressure (BP) self-screening kiosk and agreed to data transfer

|  | Practice A (n=60) | Practice B (n=126) |
|---|---|---|
| N (%) female | 34/60 (56.7) | 59 (47.2) |
| Mean (SD) age, years | 52.57 (16.8) | 50.1 (14.8) |
| Average no of kiosk sessions per user* | 1.4 | 1.3 |
| N (%) with prior diagnosis of hypertension | 13 (21.6) | 15 (11.9) |
| N (%) screening positive on kiosk who subsequently attended a consultation†‡ | 21/26 (80.8) | 18/19 (94.7) |
| N (%) screening positive on kiosk who had a prior diagnosis of hypertension† | 7/26 (26.9) | 3/19 (15.8) |
| N (%) screening positive on kiosk who subsequently underwent assessment for hypertension using home or ambulatory BP† ‡ | 0/26 (0.0) | 12/19 (63.2) |
| N (%) with a practice measurement of BP during the intervention period | 32 (53.3) | 26 (20.6)§ |
| N (%) with a practice measurement of BP during the intervention period before the first kiosk measurement | 23 (38.3) | 0 (0) |
| N (%) with a practice measurement of BP during the intervention period after the first kiosk measurement | 25 (41.7) | 26 (20.6) |

*Multiple readings on a single day are counted as a single session.
†Positive screening defined as a systolic reading of over 135 mm Hg and/or diastolic reading of over 85 mm Hg.
‡Subsequently defined as on any day after the positive screening was taken on.
§Measurements taken on the same date as kiosk measurements were excluded as indistinguishable from kiosk readings.

## Patient and public involvement

Two patient representatives sat on the steering committee of the research programme to which this study belonged. They commented on the research question, study design, methods and draft paper.

## RESULTS

### Quantitative results

We approached three pharmacy chains (representing over 4000 pharmacies), but no pharmacies were willing to take part. Stated reasons by pharmacies for declining participation included reluctance to give up retail space to house the kiosk, inability to gain consent from senior management, potential disruption to their NHS N3 internet connection and potential loss of revenue from pharmacist-led screening activities.

Two GP surgeries agreed to take part. Practice A was in a wealthy suburban village with a list size of about 5000 patients. Practice B served a population three times larger in an ethnically diverse town. The proportions of patients aged over 45 years were 64% and 43%, respectively.

When downloading the anonymised data from practice B, some technical difficulties were encountered associated with the SystmOne electronic medical record system. This limited the amount and type of data retrievable at any given time. As a result, some of the planned variables were not extracted at this site. In addition, some extracted data from this practice was found to be of questionable accuracy, which limited the analyses that could be carried out. For example, when extracting the BP readings made during the intervention period at practice B, it was not possible to extract any data that would allow identification of which readings had been automatically transferred by the kiosk.

We were able to extract these readings separately using a specific search, which permitted analysis within the kiosk readings. However, as it was not possible to know which BP measurements on a date where a kiosk reading was known to have occurred were kiosk measurements, and which were practice measurements, we limited analysis of practice measurements of BP at practice B to those taken on dates when there was no kiosk reading for that patient.

### Use of kiosks

Use of the kiosks at both GP sites was very limited. Only 60 patients (1.2%) of the eligible practice population at practice A, and 126 patients (1.2%) of the eligible population at practice B had at least one kiosk measurement successfully transferred to their electronic medical records. This included patients with existing hypertension diagnoses (13 at practice A and 15 at practice B) who were presumably using the kiosk for monitoring purposes, as well as 'true' screening by patients without a pre-existing diagnosis of hypertension.

No data were available regarding how many people used the kiosks but cancelled the session before transferring their data, or chose not to transfer their data, or gave readings to reception for manual entry. However, monitoring data were available from the supplier (table 1) showing that for a number of months during the study period, no results were transferred. This was particularly noticeable towards the end of the study period at both sites, possibly indicating that the kiosks were either malfunctioning or that there was an issue with transfer of data.

### Characteristics of kiosk users

Table 2 describes the characteristics of the patients who used the BP self-screening kiosks and agreed to

**Table 3** Impact of self-screening kiosks on hypertension detection and general practitioner practice workloads

| | Practice A | | Practice B | |
| --- | --- | --- | --- | --- |
| Period | Control | Intervention | Control | Intervention |
| Eligible population (registered, aged >18 years) | 5008 | 5137 | 9665 | 10 487 |
| N (%) of eligible population using kiosk | - | 60 (1.2) | - | 126 (1.2) |
| N (%) of new hypertensive patients detected | 50 (1.0) | 41 (0.8) | 33 (0.3) | 14 (0.1) |
| N (%) of new hypertensive patients who used the kiosk | - | 6 (10.0) | - | 0 (0.0) |
| N (%) of patients receiving blood pressure measurement by clinicians | 1810 (36.1) | 1652 (32.2) | 850 (8.8) | 1593 (15.2) |

All kiosk usage refers to use where patients agreed to data transfer

the electronic transfer of the results into their medical records (see online supplementary table 2 for details of their ethnicity). At both sites, people with and without an existing hypertension diagnosis used the kiosk as both a screening and monitoring device. The mean number of sessions per user suggests that some were using the device for repeated measures (ie, monitoring) over time.

Only 45 users across both sites had a high kiosk reading (>135 mm Hg and/or >85 mm Hg). These patients were likely to subsequently attend a consultation. Unfortunately, we have no knowledge of whether hypertension was discussed or whether the kiosk result was accessed by the treating clinician, although it would have been available to them on the patient's electronic medical record. The two sites showed a very different likelihood of subsequent diagnostic assessment for hypertension by home or ambulatory monitoring following an elevated kiosk reading. This may be related to the locally used methods of diagnosis: practice A had low use of out-of-office monitoring across both the control and intervention periods (see online supplementary table 3). At both sites, there was a higher proportion of patients with a (non-kiosk) practice measurement of BP recorded in the period after their first kiosk measurement than in the period before their first kiosk measurement. However, this increase was only significant at site B. At site A, the proportion increased by 3.3% (95% CI 16% decrease to 23% increase, p=0.85), whereas at site B, the proportion increased by 20.6% (95% CI 13% increase to 28% increase, p<0.001).

### Practice-level results

The number of patients with successful transfer of kiosk data into their medical records represented 1.2% (95% CI 1.0% to 1.4%) of the practice population at each site (table 3). Between control and intervention periods, the proportion of patients receiving BP measurement by clinicians changed in both practices, although not in a consistent direction. In practice A, there was a statistically significant 4.0% reduction in BP measurement activity (95% CI 2.1% to 5.8% decrease in activity, p<0.001). In practice B, there was a statistically significant 6.4% increase in clinician BP measurement activity (95% CI 5.5% to 7.3% increase in activity, p<0.001).

### Economic results

The cost of the hire of a kiosk for a year was £4000. Extrapolating from data on registered patients and consultations with BP measurement from the control period in practice A, it was assumed that screening would be undertaken by a combination of GPs (72.0%) and practice nurses (28.0%). For the kiosk to be cost neutral (ie, equivalent to the cost of GPs/nurses doing the same activity) at least 52.6% of patients would need to self-screen, compared with the 1.2% who actually used the kiosk.

At the level of use recorded in this study, a kiosk would cost a practice an additional £3909 a year. When comparing the cost of 100% self-screening with practice nurse only screening or HCNA only screening, the kiosk would cost a practice an additional £2123 or £2972 a year, respectively. These costs do not include the additional use of the kiosk for BP monitoring in those already diagnosed with hypertension. That would require an additional 607 or 1564 patients to use the kiosk (depending on the comparator—nurse or HCNA, respectively) for monitoring, in order to break even.

### Qualitative results

Table 4 describes the interviewee characteristics: Nine interviews were conducted with GPs and practice staff (5, 55.6% were female) while 29 interviewees were patients

**Table 4** Characteristics of Interviewees

| | Practice A | Practice B |
| --- | --- | --- |
| General practitioner (GPs) and practice Staff (n=9) | 3 GPs (2 partners, 1 salaried), 1 practice nurse, 1 member of reception/management team | 2 GPs (both partners), 1 healthcare assistant, 1 member of reception/management team |
| Patients with hypertension (n=29) | 13 (6, 46.2% had used the kiosk) | 16 (5, 31.3% had used the kiosk) |

(15, 51.7% were female). Three themes from the data are described: The 'utilisation' theme presented below explores the quantitative usage data and its capture. The 'location and ownership' and 'impact' themes offer insights into the systems' day-to-day (non-) functioning, providing some insight into the quantitative findings.

## Utilisation

The qualitative data provided a broader, complementary view[19] to the quantitative evaluation which relied on only data automatically placed in the medical records by the self-screening system. The latter could not quantify the number of patients that measured their BP in the waiting room that were (1) unable or (2) unwilling to transfer their data into their medical records. For example, although the quantitative search did not identify any patients who were detected as hypertensive via the kiosk at practice B, the interview sample did include someone who met this criteria: 'I was coming out from seeing the physio so, and I just saw it there and I thought 'well I'll, I might as well do it' (patient ID22, practice B). His kiosk reading was elevated and he was subsequently diagnosed as hypertensive.

Converging with the data provided by the system provider (table 1), the interviews demonstrated that the intended functioning of the system was intermittent. Two issues were highlighted: First, the touch screen entered a hibernation mode if not being used. Being presented with a black—apparently broken—screen deterred use. Second, there were periods when the system was offline and unable to connect to the electronic medical records. Patients would follow the onscreen prompts to enter their personal details to be told that their medical record could not be identified: 'The first time I found it really good…I managed to get all the information typed in properly and it took the reading very well, did what they said and that was great. But this time round it wouldn't take my date of birth, it wouldn't take who I was' (patient ID17, practice B).

The design of the system was not fully integrated meaning that patients could use the BP monitor without using the touch screen. Some preferred this—it was faster and enabled them to take a single measurement instead of the three instructed by the touchscreen. As one GP described: 'it's quite a lot of steps to go through to get to the point where actually you measure it and as I recall it's not terribly clear that you have to do all three readings' (GP 1, practice A).

The solid-cuff monitor seemed well liked among the interviewees that had used it. Some elderly patients however were felt to struggle with the touch screen: 'You'd say, 'It's really easy, just it'll ask you the questions, you just go ahead', and they'd be back at the desk saying, 'Not working, not finding me,' and they'd just put something in wrong that was all, but it's just older people and IT.' (Reception team member, practice B). This may, in part, account for the relatively young age of kiosk users among the hypertensive population.

If unable or unwilling to use the electronic transfer, patients could hand their results into reception: 'Some patients have mentioned that they've not been able to put their details in sometimes, but I just told them that you can still do it, press the button, it'll give you the slip, and then you can hand it in to reception, so it shouldn't necessarily be a hindrance, although it's not ideal' (GP 5, practice B). Often there were queues to speak to a receptionist at practice B which was invariably described as 'very busy' by staff and patients alike.

Having two parallel systems for results transfer presented challenges for patients, staff and the research team. It meant that duplicate instructions were needed: 'A lady stopped me and said, 'Oh, please could you show me how to use this machine because it's saying I've got to put my date of birth?' So of course the screen's not on but the piece of paper is saying, 'You've got to put your date of birth' (Practice nurse, practice A). Ad hoc procedures for manually entering and labelling these BP measurements in the electronic medical record emerged: 'Initially I think everybody was using all different codes to record it under' (Reception team member, practice A). This made it difficult to internally audit kiosk use and also meant that this data was not included in the evaluation.

While the utilisation data presented here represent the lowest estimate of the kiosks' use, they were all that was identifiable and auditable. Given the undemonstrated accuracy of BP measured in the waiting room compared that taken by a healthcare professional, its labelling and seamless inclusion in the medical record perhaps needs further thought: 'I've seen several [waiting room] readings… and the first reading could be like 188/120 or something and then the third reading is like normotensive, so I personally think that if every reading just got chucked straight into the patient records is not ideal' (GP 4, practice B).

## Location and ownership

Each practice had to locate the self-screening kiosk within the constraints of available space. Practice A opted for a lobby, prior to entering the main waiting room, as this was felt to offer improved privacy. Several patients commented the kiosk was just something you walked passed on your way into the main surgery. At practice B, the kiosk was in the far corner of the waiting room: ' It's always quite difficult to get to because the chairs are sort of there and if there's lots of people sitting there you have to ask them to move or not bother' (patient ID30, Practice B)

In both practices, the equipment was felt to be in the domain of the reception team, for whom trying to resolve problems generated additional workload: 'The only thing that I found a bit of a bother with it is that when it goes down sometime is that when I phone the helpline number, they've said their end they can see nothing wrong with it, and I'm not really good with computers… so you think you're just doing something wrong yourself' (Reception team member, practice B). The IT skills needed were beyond their usual responsibilities: 'A couple of times

the practice manager did it [reset the system], and he worked it out, obviously he's very technical. I personally didn't have a clue' (Reception team member, practice A). At both practices, the kiosk was placed behind a privacy screen preventing receptionists from being able to see if the touchscreen was working from their desk. Despite paying for a support contract, the system providers did not contact the practices to investigate periods of non-usage or the system being offline. At the same time, with the manual system for results transfer working on a-day-to-day basis, there was little motivation for practice staff to get the delicate electronic system working again.

## Impact

Unsurprisingly, given the low usage observed, clinical staff did not notice an impact on their workload. Furthermore, GPs felt that if it did improve hypertension detection, this would generate additional—although important—work.

Encouraging patients to use the system and release any downstream benefits required investment by the practice: 'I try and champion it to patients…and if they've come in for a blood pressure check, I say, 'Oh, did you know we have a machine in the waiting room where you can get it checked?' We get a mixed response from that, some are quite enthusiastic and want to try it, and others, they're not so keen' (GP 4, practice B). At practice A—located in a smaller building, where clinicians collected patients for their appointments in person, it was more feasible to demonstrate to patients how to use the system. Practice B was initially proactive in promoting the system remotely via text message and letter: 'I got a letter from the doctor to say, 'we haven't recently had a blood pressure monitor from you for a while. Would you like to just come in and do a test? You don't have to see the GP' (patient ID17, practice B). However, following the departure of the practice manager—who acted as this service champion—this did not continue. At both practices, GPs reflected that they could have done more to promote the use of the kiosk but this was difficult day to day.

## DISCUSSION

Few studies have previously evaluated the impact of BP self-screening on hypertension diagnosis rates.[11] In this study, we evaluated the feasibility of using a kiosk linked to the electronic medical records system in order to address this knowledge gap. We found that the provision of kiosks for BP measurement in practice waiting rooms resulted in few patients (1.2%) with self-screened BP measurements in their medical records. This was considerably less than previously reported screening rates in a systematic review of community-based hypertension screening.[11] This led to self-screening being more expensive for GP practices than their current provision of care. While checking BP in the waiting room was broadly found to be an appropriate and acceptable activity, the electronic transfer system was not robust or effective at capturing the BP data. Given that the latter constituted the vast majority of the cost, such a

kiosk cannot be currently recommended. Further experimental work, such as a trial, is unlikely to be worthwhile without considerable changes to either increase patient usage or improve the technical functionality.

The mixed-methods nature of this study was a major strength, providing quantitative data on the performance of the kiosks and their cost-effectiveness, and qualitative interviews to identify the strengths and weaknesses of the system and how it fitted into the workflow of a GP surgery. We were able to replicate real-life implementation of an existing commercial system in two geographically separated NHS GP practices with different populations, and using different electronic health records, ensuring that any results were not limited to a specific UK population. However, this was a small scale feasibility study, and was limited to the UK GP setting. While we intended to recruit pharmacies to the study, we were unable to find a viable business model for pharmacy BP kiosks in the UK market. In addition, significant technical issues with the automated data transfer system limited the data available.

While manufacturers and retailers emphasise the time-saving benefits, GP interviewees suggested that increased self-screening could lead to improved detection and additional hypertension management workload. Furthermore, the screening accuracy of such systems remains unevaluated and therefore it is not yet possible to provide thresholds of self-screened BP to rule out hypertension, thus avoiding rechecking by healthcare professionals, although these are likely to be similar to those for ambulatory monitoring.[21] In our study, we observed increased levels of GP workload at one site, and decreased levels at the other, indicating that any benefits may be contingent.

Simple cost analysis showed that, at the rate of usage and the size of practice in this evaluation, this kiosk would not be cost-effective. A simpler, cheaper system using administrative staff to enter the data instead of the computer link may be more appropriate for practices who wish to offer self-screening. We observed this was a frequent work around for technical problems with the kiosk. Previous research reported that these simpler systems were felt by primary care staff to be helpful in attaining performance targets and generally acceptable to both patients and staff.[22–24] However, while Hamilton *et al* successfully recruited community sites to host standalone BP monitors in a study of hypertension self-screening, many users did not supply their hand-written results slip to their GP.[25] This highlights the difficulties of ensuring BP self-screening follow-up, whether 'high tech' or 'low tech' systems are used.

Our results suggest that a kiosk located at the surgery does not necessarily extend the screening reach of practices. Future work could investigate how to promote self-screening in order to increase usage particularly among groups less likely to have their BP checked. For example, men are known to have lower consultation rates than women.[26] However, a very significant increase in usage from such activities would be required simply to avoid a loss of reach compared with current provisions

of care. We were unable to quantify the extent to which other methods of self-screening, such as use of the kiosk without automated transfer or home measurement of BP, were used during the study.

While novel technologies can be very appealing,[27] their adoption into daily practice can be challenging.[28] For example, the reception teams in our study found the requirements of the automated transfer associated with the kiosk to be beyond their typical responsibilities and, in some cases, expertise. Greenhalgh *et al* published a framework to help predict the success of healthcare technologies.[29] The kiosk used in this study scores 'complicated' under several domains: first, the technology is not 100% reliable with the manufacturer's claims about data capture not being achieved. We are aware of other BP self-screening systems with automated data transfer technology experiencing similar problems that also only become apparent on data extraction (McManus, personal communication). Second, the value proposition is unclear with the very poor efficacy and cost-effectiveness in this feasibility work. Third, as noted above, existing staff must learn new skills or new staff hired which may counter balance any savings even in a well-used service. Finally, there is the restricted capacity of GP practices to change given their limited resources, variable leadership and managerial relations. Together these suggest that further work is needed prior to considering the widespread introduction of BP self-screening systems linked to the practice electronic medical records in primary care. Providing healthcare commissioners and GP practices with information such as this will help them make informed decisions when contemplating the charms of new technologies in an environment of resource constraint and opportunity costs.

## CONCLUSIONS

These results indicate that a larger scale trial using the same methods and equipment is not, at this stage, feasible, nor can the self-screening equipment with automated data transfer tested in this study be recommended for use in primary care. The findings also highlight uncertainty about the robustness of such a system for use in daily clinical practice.

**Acknowledgements** The research team are very grateful to the GP surgeries and interviewees that participated in this study. They also acknowledge the valuable input of David Yeomans and Derek Shaw, the research programme's PPI representatives.

**Contributors** AW, CH and RJM designed the study. SF and AT collected the data with support from DM. SF and AT led the analyses and drafted the manuscript. M-ML contributed to the quantitative analysis. MM and SJ undertook the economic analysis. All authors, including SG and RH, commented on and approved the manuscript. RJM will be the guarantor.

**Funding** This article presents independent research funded by a National Institute for Health Research Programme Grant RP-PG-1209–10051. RJM was supported by an NIHR professorship (NIHR-RP-02-12-015) and now by the NIHR Collaboration for Leadership in Applied Health Research and Care (CLAHRC) Oxford at Oxford Health NHS Foundation Trust. CH is partly supported by the NIHR Biomedical Research Centre, Oxford and by the NIHR School of Primary Care Research. RH acknowledges part support from the NIHR School for Primary Care Research (SPCR), the NIHR Collaboration for Leadership in Applied Research in Health and Care (CLARHC) Oxford and the NIHR Biomedical Research Centre (BRC), Oxford. SG is partly funded by the National Institute for Health Research (NIHR) Collaboration for Leadership in Applied Health Research and Care West Midlands. DM was supported by an NIHR In Practice Fellowship at the time of the work.

**Disclaimer** The views expressed are those of the author(s) and not necessarily those of the NHS, the NIHR or the Department of Health.

**Competing interests** SF, AT, M-ML, SJ, MM, SG, RH, AW, DM and RJM declare no competing interests. CH has received expenses and payments for media work. He has received expenses from WHO and holds grant funding from the NIHR, the NIHR School of Primary Care Research, The Wellcome Trust and WHO. On occasion, he receives expenses for teaching EBM and is an NHS GP in the out of hours service in Oxford.

**Patient consent for publication** Not required.

**Ethics approval** The study (qualitative and quantitative components) was approved by NRES South West (Cornwall and Plymouth) Research Ethics Committee (14/SW/1167).

**Provenance and peer review** Not commissioned; externally peer reviewed.

**Data sharing statement** No additional data are available.

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
