## [Reviewer comments · BMJ Open]

ARTICLE DETAILS

TITLE (PROVISIONAL)	A mixed-methods feasibility study of blood pressure self-screening for hypertension detection
AUTHORS	Tompson, Alice; Fleming, Susannah; Lee, Mei-Man; Monahan, Mark; Jowett, Sue; McCartney, David; Greenfield, Sheila; Heneghan, Carl; Ward, Alison; Hobbs, Richard; McManus, Richard

VERSION 1 - REVIEW

REVIEWER	Stefano Omboni Italian Institute of Telemedicine Italy
REVIEW RETURNED	30-Nov-2018

GENERAL COMMENTS	In this feasibility study of a BP self-measurement kiosk authors conclude that (abstract) "Whilst interviewees were generally positive about checking BP in the waiting room, the electronic transfer system was neither robust, effective nor likely to be a cost-effective approach. Since most of the cost of a kiosk system lies in the transfer mechanism, a simpler system using a solid cuff sphygmomanometer and manual entry of results may be currently more appropriate for the primary care environment" and that (conclusions) "Our results suggest that a kiosk located at the surgery does not extend the screening reach of practices... We were unable to quantify the extent to which other methods of self-screening, such as use of the kiosk without automated transfer or home measurement of blood pressure, were used during the study". I do not agree with the conclusions, except that the place for this solution could be somewhere else than GPs' practices. Feasibility of a technology and its acceptability depend on the appropriateness and usability of the technology / model proposed. Allegedly, authors selected the wrong technology and the wrong model, so concluding that solid-cuff sphygmomanometers + manual entry are the best approaches is not the correct conclusion. Many automatic kiosks are available worldwide mainly in pharmacies. Most of them need some help from the operator, but they allow automatic data transfer and storage and are very practical. I recommend to discuss this aspect and compare the present feasibility study with those of other technologies which indeed worked.
---

	I understand the technology used in this study is not disruptive. Indeed such approaches with proper technology are disruptive. They may also be cost-effective. Maybe this should be discussed. Why did the authors embark themselves in such a study involving practices and patients without making a simple internal feasibility study? This would have helped to refine the technology and model. Please discuss. Why the 135/85 mmHg threshold was used for triggering the identification of high BP levels. Is that choice based on proposed unattended BP thresholds? Please, support with references. The disappointedly low use of the kiosk by patients is not only a matter of bad technology in this study. This is what regularly happens with patients' screening in the community. Patients do not want to check themselves unless they know already to have high BP. Please discuss. This study lacks of a quantitative acceptability analysis. This would have helped instead of a qualitative analysis.
--	---

REVIEWER	Jordana Cohen, MD, MSCE Renal-Electrolyte and Hypertension Division and Clinical Center for Epidemiology and Biostatistics University of Pennsylvania, USA
REVIEW RETURNED	10-Dec-2018

GENERAL COMMENTS	This is a study evaluating the feasibility kiosk-mediated self-measurement of blood pressure for the purpose of screening for hypertension. Given increasing evidence of the inadequacy of in-office blood pressure measurement for the screening and diagnosis of hypertension, access to and feasibility of more accurate screening measures for hypertension are timely and important issues. However, I have several concerns regarding the approach and discussion: 1) There was a prolonged period of time that the authors note no readings were transferred to the EHR, potentially due to technical difficulties. The authors approach this as indication of inadequacy of the device. I think a lot can be learned regarding future use of these devices with regard to troubleshooting and procedural planning. I am surprised that there was no component of the protocol included to check intermittently to see if the devices were working. The inability to assess if there was a functional transfer link unfortunately makes the first aim of the study (i.e. the number of patients using the kiosk and agreeing to transfer their data) uninterpretable/invalidated. Many electronic devices link to EHR's in other contexts in which this type of a technical issue is not considered an acceptable barrier. Pilot testing would also have potentially addressed this issue prior to undertaking the study. I recommend that the authors consider approaching this as a limitation of the study rather than as an indication that the intervention was not viable. 2) In the discussion, the authors criticize kiosks as not being time saving and indicate that the accuracy of these kiosks has not been evaluated. I ask the authors to rethink this paragraph and the tone of the discussion. For example, please discuss your findings in the
--

	context of existing qualitative studies that found differing results, such as Chung et al. J Am Board Fam Med 2016 Sep-Oct;29(5)620-9. Additionally, kiosk measurement, much like AOBP, is much more accurate than in-office measurement. There are existing kiosk studies that demonstrate a close correlation of kiosk measurement to daytime ABPM (for example, Padwal et al. J Am Soc Hypertens. 2015 Feb;9(2):123-9), which has the potential to improve the ability to more accurately identify individuals with undiagnosed hypertension. 3) It seems that the kiosks captured many patients who had positive screenings (30% of individuals in Practice A and 15% in Practice B). The authors call this “a small number of patients” in the results section. Was there a predefined threshold for the number of patients needed to have a positive screening for the kiosks to be determined an acceptable alternative to an in-office blood pressure? I recommend that the authors change the wording to be more specific in the results section (as “a small number” seems like a value judgment). This also merits inclusion in the discussion. 4) I encourage the authors to perform a cost-benefit analysis that also takes into account the improved ability of kiosks to overcome white coat effect. Given that evidence that suggests that kiosks are more accurate than a clinic blood pressure measurement, it seems that the comparison of cost between the average of 3 kiosk measurements vs. a single in-office measurement is somewhat incongruous. Since kiosk measurements more closely estimate mean daytime ambulatory monitoring than in-office measurement (and both tend to overcome masked hypertension), this seems like a potentially appropriate secondary method for comparison. 5) Was there any assessment of whether the patients were using the kiosk correctly beyond the transferring of their data to the electronic medical record? 6) Could the authors please specify if the kiosks used in the study are validated? Up to what arm circumference are they able to support? 7) The authors state in the results that “no pharmacies were willing to take part” – please include the number of pharmacies and GP surgeries that were approached
--	---

VERSION 1 – AUTHOR RESPONSE

Reviewer: 1

Reviewer Name: Stefano Omboni

Institution and Country: Italian Institute of Telemedicine Italy

Please state any competing interests or state 'None declared': None declared

Please leave your comments for the authors below

In this feasibility study of a BP self-measurement kiosk authors conclude that (abstract) “Whilst interviewees were generally positive about checking BP in the waiting room, the electronic transfer system was neither robust, effective nor likely to be a cost-effective approach. Since most of the cost of a kiosk system lies in the transfer mechanism, a simpler system using a solid cuff sphygmomanometer and manual entry of results may be currently more appropriate for the primary care environment” and that (conclusions) “Our results suggest that a kiosk located at the surgery does not extend the screening reach of practices... We were unable to quantify the extent to which other methods of self-screening, such as use of the kiosk without automated transfer or home measurement of blood pressure, were used during the study”.

I do not agree with the conclusions, except that the place for this solution could be somewhere else than GPs' practices. Feasibility of a technology and its acceptability depend on the appropriateness and usability of the technology / model proposed. Allegedly, authors selected the wrong technology and the wrong model, so concluding that solid-cuff sphygmomanometers + manual entry are the best approaches is not the correct conclusion. Many automatic kiosks are available worldwide mainly in pharmacies. Most of them need some help from the operator, but they allow automatic data transfer and storage and are very practical. I recommend to discuss this aspect and compare the present feasibility study with those of other technologies which indeed worked.

I understand the technology used in this study is not disruptive. Indeed such approaches with proper technology are disruptive. They may also be cost-effective. Maybe this should be discussed.

Response:

We do not believe we selected the wrong technology to trial. The blood pressure monitor used in this study was a standard monitor, which is both clinically validated, and widely used for self-monitoring of blood pressure by patients in clinical settings (TM-2655P, A&D Instruments Ltd, UK). The kiosk technology chosen to allow the data from the blood pressure monitor to be automatically transferred to the electronic health record was the only one that we were able to find that was compatible with a high proportion (more than 80%) of electronic health record systems in use in UK primary care. The UK NHS is a unique setting with a single primary care patient electronic health record, held by the patient's general practitioner.

While it is theoretically possible for pharmacy-located kiosks to access this record, there are additional technical and regulatory barriers involved in doing so, that may not be present in other health systems. We did however wish to assess a UK experience of BP measurement kiosks which the reviewer correctly points out are available elsewhere. Our initial work in this study (reference 22 in the paper) found no pharmacies locally had such equipment. We therefore approached 2 large national chains of pharmacies and one local chain, representing a total of over 4,000 pharmacies, to attempt to recruit pharmacy sites to the study. However, the responses we received were that, while UK pharmacies do professional-led BP monitoring, they have no business case for the introduction of kiosks, and are unwilling to devote potentially profitable retail floor to a device that has no, or only limited, associated income stream. Again, this tallied with our previous work on blood pressure screening in UK pharmacies (reference 22 in the paper.)

Why did the authors embark themselves in such a study involving practices and patients without making a simple internal feasibility study? This would have helped to refine the technology and model. Please discuss.

Response:

The submitted paper is describing such a feasibility study. Since the majority of the disruption associated with the intervention occurred during installation of the equipment, we did not consider that

a shorter or more limited feasibility study would be of any additional benefit. Choice of technology was limited by our requirement for an existing kiosk device that could interface with UK National Health Service electronic health records, and that took seated blood pressure using a validated device. We have added the following to the start of the methods section to clarify our choice of technology:

“We selected a BP self-measurement kiosk device (SurgeryPod Plus, Medvivo latterly Microtech, UK) that was able to interface with various UK NHS general practice electronic health record systems, and which took seated blood pressure using an clinically validated sphygmomanometer.”

Why the 135/85 mmHg threshold was used for triggering the identification of high BP levels. Is that choice based on proposed unattended BP thresholds? Please, support with references.

Response:

The threshold was chosen based on the UK NICE guidelines for clinical management of primary hypertension in adults (reference 10 in the paper) and is similar to thresholds suggested for unattended automated blood pressure. Such thresholds are not commonplace in the UK and so we chose to match the clinical threshold for home and ambulatory blood pressure, as we felt that self-measured kiosk blood pressure was more similar to these measures than to clinic blood pressure, which is more likely to be subject to white coat effects. We have added the following text to end of the second paragraph of the Methods section:

“The trigger level was chosen to match the clinical threshold for home and ambulatory blood pressure measurement in the UK NICE guidelines.(10)”

The disappointedly low use of the kiosk by patients is not only a matter of bad technology in this study. This is what regularly happens with patients’ screening in the community. Patients do not want to check themselves unless they know already to have high BP. Please discuss.

Response:

We agree that low screening rates are expected from self-screening, and that a considerable proportion of self-screener might be expected to have an existing hypertension diagnosis. However, the usage seen in this study was considerably lower than any reported in our previous systematic review of self-screening and non-physician screening for hypertension (reference 11). We have added the following text to the first paragraph of the discussion:

“This was considerably less than previously reported screening rates in a systematic review of community-based hypertension screening.(11)”

This study lacks of a quantitative acceptability analysis. This would have helped instead of a qualitative analysis.

Response:

We did not carry out a quantitative acceptability assessment, as we felt that the information gathered from in-depth interviews with both practice staff and patients (including both patients who used the kiosk device, and those who chose not to), would give a much more detailed picture of the impact of

the kiosk device. The interviews also allowed us to identify the reasons underlying the attitudes to the kiosk device, which cannot be extracted from a quantitative assessment of acceptability.

Reviewer: 2

Reviewer Name: Jordana Cohen, MD, MSCE

Institution and Country: Renal-Electrolyte and Hypertension Division and Clinical Center for Epidemiology and Biostatistics, University of Pennsylvania, USA

Please state any competing interests or state 'None declared': None

Please leave your comments for the authors below

This is a study evaluating the feasibility kiosk-mediated self-measurement of blood pressure for the purpose of screening for hypertension. Given increasing evidence of the inadequacy of in-office blood pressure measurement for the screening and diagnosis of hypertension, access to and feasibility of more accurate screening measures for hypertension are timely and important issues. However, I have several concerns regarding the approach and discussion:

1) There was a prolonged period of time that the authors note no readings were transferred to the EHR, potentially due to technical difficulties. The authors approach this as indication of inadequacy of the device. I think a lot can be learned regarding future use of these devices with regard to troubleshooting and procedural planning. I am surprised that there was no component of the protocol included to check intermittently to see if the devices were working. The inability to assess if there was a functional transfer link unfortunately makes the first aim of the study (i.e. the number of patients using the kiosk and agreeing to transfer their data) uninterpretable/invalidated. Many electronic devices link to EHR's in other contexts in which this type of a technical issue is not considered an acceptable barrier. Pilot testing would also have potentially addressed this issue prior to undertaking the study. I recommend that the authors consider approaching this as a limitation of the study rather than as an indication that the intervention was not viable.

Response:

We agree that the loss of transfer during the study is a limitation of the study, although our intention was to see if kiosks with automated transfer would be suitable for introduction into the normal workflow of a GP surgery. The only way of assessing whether the transfer link was active was by monitoring the number of transferred sessions (Table 1) – this information was only available to the kiosk supplier (not directly to the practice or the researchers), and was only provided to the researchers when we specifically queried the suppliers about it.

We did not do this systematically through the study as we aimed to assess the real life implementation of an existing commercially available self-screening solution in UK Primary Care. As such, we felt that intervention by the research team, whilst potentially troubleshooting issues, would have influenced the results away from routine implementation.

As noted in the qualitative results (section headed "Location and Ownership"), despite the kiosk supplier having access to the usage data, they made no effort to contact either the researchers or the practices to investigate this, and a number of practice staff experienced considerable difficulties in resetting or troubleshooting the system when it did have technical issues.

These issues may be limited to this particular kiosk device / manufacturer and may not be generalizable to all similar devices, but we feel it is important to highlight them.

We have added a new paragraph on the strengths and limitations of the study to the Discussion section to address this and other issues:

“The mixed methods nature of this study was a major strength, providing quantitative data on the performance of the kiosks and their cost-effectiveness, and qualitative interviews to identify the strengths and weaknesses of the system and how it fitted into the workflow of a GP surgery. We were able to replicate real-life implementation of an existing commercial system in two geographically separated NHS GP practices with different populations, and using different electronic health records, ensuring that any results were not limited to a specific UK population. However, this was a small scale feasibility study, and was limited to the UK GP setting. While we intended to recruit pharmacies to the study, we were unable to find a viable business model for pharmacy BP kiosks in the UK market. In addition, significant technical issues with the automated data transfer system limited the data available.”

2) In the discussion, the authors criticize kiosks as not being time saving and indicate that the accuracy of these kiosks has not been evaluated. I ask the authors to rethink this paragraph and the tone of the discussion. For example, please discuss your findings in the context of existing qualitative studies that found differing results, such as Chung et al. *J Am Board Fam Med* 2016 Sep-Oct;29(5):620-9. Additionally, kiosk measurement, much like AOBP, is much more accurate than in-office measurement. There are existing kiosk studies that demonstrate a close correlation of kiosk measurement to daytime ABPM (for example, Padwal et al. *J Am Soc Hypertens*. 2015 Feb;9(2):123-9), which has the potential to improve the ability to more accurately identify individuals with undiagnosed hypertension.

Response:

We agree that previous qualitative studies investigating kiosk devices using paper slips rather than automated transfer, including the paper by Chung et al, have found that both patients and staff find these devices acceptable. This is why we believe that such devices are likely to be more appropriate for use in primary care than the kiosk type used in this study, since it was the automated transfer system that appeared to be the root of the issues experienced by staff and patients in our study and hence our conclusion regarding the use of solid cuff monitors and manual entry. We accept that we may not have made this sufficiently clear in the discussion or conclusions, and have made various textual changes to clarify this. We have included a citation to the suggested paper by Chung et al in the discussion (para 4):

“Previous research reported that these simpler systems were felt by primary care staff to be helpful in attaining performance targets and generally acceptable to both patients and staff.(22,23,24).”

We have also clarified the conclusions:

“These results indicate that a larger scale trial using the same methods and equipment is not, at this stage, feasible nor can self-screening equipment with automated data transfer be recommended.”

We agree that kiosk measurement is accurate, but screening accuracy (i.e. whether kiosk measurement correctly detects hypertension), and the appropriate thresholds to be used for ruling out hypertension using self-screened BP have not yet been determined. The data in Padwal et al gives an indication that the threshold is likely to be similar to that for ABPM, but the high variability between

the measurements and the limited population mean that this is not certain. We have included this in the discussion text (paragraph 3):

“Furthermore, the screening accuracy of such systems remains unevaluated and therefore it is not yet possible to provide thresholds of self-screened BP to rule out hypertension, thus avoiding rechecking by healthcare professionals, although these are likely to be similar to those for ambulatory monitoring.(21)”

3) It seems that the kiosks captured many patients who had positive screenings (30% of individuals in Practice A and 15% in Practice B). The authors call this “a small number of patients” in the results section. Was there a predefined threshold for the number of patients needed to have a positive screening for the kiosks to be determined an acceptable alternative to an in-office blood pressure? I recommend that the authors change the wording to be more specific in the results section (as “a small number” seems like a value judgment). This also merits inclusion in the discussion.

Response:

The number of patients who had a positive screening was 26 in Practice A, and 19 in practice B (Table 2.) We considered this to be a small number compared to the eligible practice population of 5137 in practice A and 10487 in Practice B (Table 3.) The percentages reported in Table 2 referred to by the reviewer are the proportion who screened positive on the kiosk and had a prior diagnosis of hypertension. It is not unexpected that many patients with positive screening were already known to be hypertensive.

There was not a pre-defined threshold for the number of patients. The intention of the text was to show that we had very little data on participants with high screening blood pressure, and so we have clarified the text to refer to the absolute number (Characteristics of kiosk users, paragraph 2):

“Only 45 users across both sites had a high kiosk reading (> 135mmHg and/or >85mmHg).”

4) I encourage the authors to perform a cost-benefit analysis that also takes into account the improved ability of kiosks to overcome white coat effect. Given that evidence that suggests that kiosks are more accurate than a clinic blood pressure measurement, it seems that the comparison of cost between the average of 3 kiosk measurements vs. a single in-office measurement is somewhat incongruous. Since kiosk measurements more closely estimate mean daytime ambulatory monitoring than in-office measurement (and both tend to overcome masked hypertension), this seems like a potentially appropriate secondary method for comparison.

Response:

We agree that kiosk blood pressure would be expected to more closely match ambulatory or home blood pressure. However, we were not able to reliably extract ambulatory blood pressure data from the electronic patient records, so it would not be possible to carry out an analysis using this as a comparator. The limited use of the automated transfer, and the paucity of data available for some extracted variables precluded the planned cost-benefit analysis.

5) Was there any assessment of whether the patients were using the kiosk correctly beyond the transferring of their data to the electronic medical record?

Response:

There was no formal assessment of this during the study as it was not feasible to carry out ethnographic assessment, and the areas chosen for the kiosks were not typically visible to practice staff for patient privacy reasons. However, we did elicit a number of issues with patients struggling to use the kiosk correctly through the qualitative interviews. These included use of the solid cuff device independently of the touchscreen, and difficulty in entering details so that they could be successfully identified on the practice system. Examples of this can be seen in the “Utilisation” subsection of the Results section. For example:

““The first time I found it really good...I managed to get all the information typed in properly and it took the reading very well, did what they said and that was great. But this time round it wouldn't take my date of birth, it wouldn't take who I was” (Patient ID17, practice B).”

““You'd say, 'It's really easy, just it'll ask you the questions, you just go ahead,' and they'd be back at the desk saying, 'Not working, not finding me,' and they'd just put something in wrong that was all, but it's just older people and IT.” (Reception team member, practice B).”

““Some patients have mentioned that they've not been able to put their details in sometimes, but I just told them that you can still do it, press the button, it'll give you the slip, and then you can hand it in to reception, so it shouldn't necessarily be a hindrance, although it's not ideal” (GP 5, practice B).”

6) Could the authors please specify if the kiosks used in the study are validated? Up to what arm circumference are they able to support?

Response:

The sphygmomanometer used in the kiosk device is clinically validated, and supports arm circumferences from 17 to 45 cm. We have added this information to the text:

Methods section, para 2:

“The kiosks combined a clinically validated solid cuff automated sphygmomanometer (TM-2655P, A&D Instruments Ltd, UK), supporting arm circumferences from 17 to 45cm, with a dedicated touch screen computer interface.(14)”

7) The authors state in the results that “no pharmacies were willing to take part” – please include the number of pharmacies and GP surgeries that were approached

Response:

We approached 2 national pharmacy chains and 1 local pharmacy chain, representing over 4,000 pharmacies nationally, without success. Due to the way GP surgeries were recruited (through the NHS clinical research network), we do not know how many were approached in order to recruit the planned 2 surgeries. There is no equivalent research infrastructure in the UK for recruiting pharmacies. We have edited the first line of the results section to read:

“We approached three pharmacy chains (representing over 4,000 pharmacies), but no pharmacies were willing to take part.”

VERSION 2 – REVIEW

REVIEWER	Stefano Omboni Italian Institute of Telemedicine, Italy
REVIEW RETURNED	19-Mar-2019

GENERAL COMMENTS	I am satisfied with the replies, except for the response to my previous comment #1. The conclusions in the abstract “Whilst interviewees were generally positive about checking BP in the waiting room, the electronic transfer system was neither robust, effective nor likely to be a cost-effective approach. Since most of the cost of a kiosk system lies in the transfer mechanism, a simpler system using a solid-cuff sphygmomanometer and manual entry of results may be currently more appropriate for the primary care environment” and main text “These results indicate that a larger scale trial using the same methods and equipment is not, at this stage, feasible nor can self-screening equipment with automated data transfer be recommended. The findings also highlight uncertainty about the robustness of such a system for use in daily clinical practice.” I do not agree with this conclusion because my experience is completely different. I recommend smoothing the conclusions, because these apply to the technology used in this study. The authors cannot exclude that other technologies/approaches may have been successful (as in my experience). I recommend rewriting the conclusions of the abstract and main text. Some suggestions. Abstract: “Whilst interviewees were generally positive about checking BP in the waiting room, the electronic transfer system was neither robust, effective nor likely to be a cost-effective approach, thus may not be currently appropriate for the primary care environment”. Main text: “These results indicate that, at this stage, a larger scale trial using the same methods and equipment might not be feasible. The findings also highlight uncertainty about the robustness of such a system for use in daily clinical practice”.
--

REVIEWER	Jordana Cohen, MD, MSCE Renal-Electrolyte and Hypertension Division and Department of Epidemiology, University of Pennsylvania, USA
REVIEW RETURNED	03-Apr-2019

GENERAL COMMENTS	The authors' changes have substantially clarified the objectives of the study, and the discussion now expresses the conclusions much more clearly and appropriately based on the results. I have no further comments.
--

VERSION 2 – AUTHOR RESPONSE

Reviewer: 1

Reviewer Name: Stefano Omboni

Institution and Country: Italian Institute of Telemedicine, Italy Please state any competing interests or state 'None declared': None declared

Please leave your comments for the authors below

I am satisfied with the replies, except for the response to my previous comment #1. The conclusions in the abstract "Whilst interviewees were generally positive about checking BP in the waiting room, the electronic transfer system was neither robust, effective nor likely to be a cost-effective approach. Since most of the cost of a kiosk system lies in the transfer mechanism, a simpler system using a solid-cuff sphygmomanometer and manual entry of results may be currently more appropriate for the primary care environment" and main text "These results indicate that a larger scale trial using the same methods and equipment is not, at this stage, feasible nor can self-screening equipment with automated data transfer be recommended. The findings also highlight uncertainty about the robustness of such a system for use in daily clinical practice." I do not agree with this conclusion because my experience is completely different. I recommend smoothing the conclusions, because these apply to the technology used in this study. The authors cannot exclude that other technologies/approaches may have been successful (as in my experience). I recommend rewriting the conclusions of the abstract and main text. Some suggestions. Abstract: "Whilst interviewees were generally positive about checking BP in the waiting room, the electronic transfer system was neither robust, effective nor likely to be a cost-effective approach, thus may not be currently appropriate for the primary care environment". Main text: "These results indicate that, at this stage, a larger scale trial using the same methods and equipment might not be feasible. The findings also highlight uncertainty about the robustness of such a system for use in daily clinical practice".

Response:

We agree that the conclusions in this paper are only applicable to the system tested, and do not intend to imply otherwise. We are aware that other technologies and approaches have been used successfully, but are not directly applicable to the UK primary care environment. We have therefore edited the final paragraph of the conclusions to read:

"Whilst interviewees were generally positive about checking BP in the waiting room, the electronic transfer system as tested was neither robust, effective nor likely to be a cost-effective approach, thus may not be appropriate for a primary care environment. Since most of the cost of a kiosk system lies in the transfer mechanism, a solid-cuff sphygmomanometer and manual entry of results may be a suitable alternative."

We have also edited the final paragraph of the Conclusions section to read:

"These results indicate that a larger scale trial using the same methods and equipment is not, at this stage, feasible, nor can the self-screening equipment with automated data transfer tested in this study be recommended for use in primary care. The findings also highlight uncertainty about the robustness of such a system for use in daily clinical practice."

Reviewer: 2

Reviewer Name: Jordana Cohen, MD, MSCE

Institution and Country: Renal-Electrolyte and Hypertension Division and Department of Epidemiology, University of Pennsylvania, USA Please state any competing interests or state 'None declared': None declared

Please leave your comments for the authors below

The authors' changes have substantially clarified the objectives of the study, and the discussion now expresses the conclusions much more clearly and appropriately based on the results. I have no further comments.